# Enhancement of Structural, Optical and Photoelectrochemical Properties of n−Cu_2_O Thin Films with K Ions Doping toward Biosensor and Solar Cell Applications

**DOI:** 10.3390/nano13071272

**Published:** 2023-04-04

**Authors:** Mahmoud Abdelfatah, Nourhan Darwesh, Mohamed A. Habib, Omar K. Alduaij, Abdelhamid El-Shaer, Walid Ismail

**Affiliations:** 1Physics Department, Faculty of Science, Kafrelsheikh University, Kafrelsheikh 33516, Egyptelshaer@sci.kfs.edu.eg (A.E.-S.);; 2Department of Chemistry, College of Science, Imam Mohammad Ibn Saud Islamic University (IMSIU), Riyadh P.O. Box 90952, Saudi Arabia; 3Chemistry of Tanning Materials and Leather Technology Department, Chemical Industries Institutes, National Research Center, Dokki, Giza P.O. Box 12622, Egypt

**Keywords:** n−Cu_2_O thin films, potassium doping, electrodeposition, biosensor, solar cell applications

## Abstract

n-type Cu_2_O thin films were grown on conductive FTO substrates using a low-cost electrodeposition method. The doping of the n−Cu_2_O thin films with K ions was well identified using XRD, Raman, SEM, EDX, UV-vis, PL, photocurrent, Mott–Schottky, and EIS measurements. The results of the XRD show the creation of cubic Cu_2_O polycrystalline and monoclinic CuO, with the crystallite sizes ranging from 55 to 25.2 nm. The Raman analysis confirmed the presence of functional groups corresponding to the Cu_2_O and CuO in the fabricated samples. Moreover, the samples’ crystallinity and morphology change with the doping concentrations which was confirmed by SEM. The PL results show two characteristic emission peaks at 520 and 690 nm which are due to the interband transitions in the Cu_2_O as well as the oxygen vacancies in the CuO, respectively. Moreover, the PL strength was quenched at higher doping concentrations which reveals that the dopant K limits e−/h+ pairs recombination by trapped electrons and holes. The optical results show that the absorption edge is positioned between 425 and 460 nm. The computed Eg for the undoped and K−doped n−Cu_2_O was observed to be between 2.39 and 2.21 eV. The photocurrent measurements displayed that the grown thin films have the characteristic behavior of n-type semiconductors. Furthermore, the photocurrent is enhanced by raising the doped concentration, where the maximum value was achieved with 0.1 M of K ions. The Mott–Schottky measurements revealed that the flat band potential and donor density vary with a doping concentration from −0.87 to −0.71 V and 1.3 × 10^17^ to 3.2 × 10^17^ cm^−3^, respectively. EIS shows that the lowest resistivity to charge transfer (Rct) was attained at a 0.1 M concentration of K ions. The outcomes indicate that doping n−Cu_2_O thin films are an excellent candidate for biosensor and photovoltaic applications.

## 1. Introduction

Copper oxides are one of the semiconductor oxides that have attracted the most attention in solar cell and sensor applications due to their extreme physicochemical characteristics [1,2]. Cu_2_O generally has the characteristic behavior of a p-type semiconductor owing to an acceptor level created by copper ion vacancies formed above the valence band at about 0.4 eV. Donor levels created by oxygen vacancies below the conduction band bottom at about 0.38 eV indicate the presence of a Cu_2_O n-type semiconductor [3]. Cu_2_O exhibits a direct band gap of about 2 eV, an abundance of source materials, non-toxicity and an extremely high absorption coefficient [4,5]. There are different techniques used to fabricate Cu_2_O, including RF magnetron sputtering [6], pulsed laser deposition [7], chemical vapor deposition [7], the solvothermal method [8], the hydrothermal method [9] atomic layer deposition (ALD) [10], sol-gel [11], and electrochemical deposition [12]. The electrodeposition technique was mainly utilized to produce the n−Cu_2_O thin film because of its low cost, and it is well done at a low temperature [13]. Cu_2_O is a promising candidate for photovoltaic energy conversion [4,5]. Although Cu_2_O solar cells have a theoretical efficiency of about 20%, an efficiency of around 9.54% was reported for Cu_2_O/Si heterojunction photovoltaics [14]. This could be attributed to many reasons. The first is the lack of n-type Cu_2_O which results in an interface between the p-type Cu_2_O and other materials and a significant loss mechanism for the separation and collecting of photo-carriers [15]. The second is the minority of the carrier concentration [16,17]. The best solution to overcome this problem is the doping process, in order to improve the carrier concentration and the optical and electronic properties of such materials, and therefore the efficiency of photo conversion could be improved [4,18]. In this work, n−Cu_2_O was potentiostatically electrodeposited on FTO conductive substrates where many optoelectronic devices such as a solar cell need front and back contacts to collect the charge carriers and therefore the FTO could be used as an electrode for devices. The doping process is introduced to improve the properties of n−Cu_2_O where the effect of the K ions doping at various concentrations (0, 0.03, 0.05, 0.07 and 0.1 M) on the microstructural, morphological, optical and photoelectrochemical Cu_2_O properties was investigated deeply.

## 2. Materials and Methods

In our work, pure and K−doped n−Cu_2_O thin films were deposited at different concentrations (0, 0.03, 0.05, 0.07 and 0.1 M) on 1 cm × 1 cm glass substrates coated with FTO (lower than 12 Ohm/Sq, Sigma Aldrich, St. Louis, MO, USA) using electrochemical deposition technique with three-electrodes system [19]. FTO substrates were cleaned before electrodeposition with ethanol, acetone and deionized water (DI) for 20 min, respectively. The solution consists of 0.45 M of CuSO_4_.5H_2_O (copper sulfate pentahydrate, purity ≥ 99%, Merck, Darmstadt, Germany) as copper ions source, and 3 M of Lactic acid (Merck, purity of about 90%) as a stabilizer agent. The solution was stirred for 20 min and then 4 M of sodium hydroxide was added to the solution (NaOH, with purity ≥ 98%, Merck) to adjust the value of pH at 6.8. For doping with K ions source, the potassium sulfate (K_2_SO_4_, purity ≥ 99%, Merck) was added to the solution at various concentrations (0, 0.03, 0.05, 0.07 and 0.1 M). Bio-logic SAS mode1: SP-50 s/n 0092 was employed at potentiostatic mode to grow undoped and K−doped n−Cu_2_O thin films on FTO substrates at a constant potential of −0.4 V and constant temperature 60 °C for 5 min. The obtained samples were rinsed with DI followed by drying in the oven. The structure and morphology of fabricated thin films were characterized employing X-ray diffraction (XRD-6000 Shimadzu) in addition to scanning electron microscope (SEM) (JSM-651OLV). UV-vis spectrophotometer (JASCO V-630) and a Kimmon He-Cd laser with a HORIBA iHR320 spectrometer were applied to examine the optical absorption and produce photoluminescence, respectively. Bio-LogicSb-50 potentiostat system with three electrodes was utilized to determine the photoelectrochemical properties of all fabricated samples. Mott–Schottky and Electrochemical Impedance Spectroscopy (EIS) Measurements were performed employing CHI660E electrochemical workstation.

## 3. Results and Discussion

### 3.1. Structural Investigations

Figure 1 represents the XRD patterns for the pure and K−doped Cu_2_O thin films as the n-type deposited on an FTO substrate. The fabricated films are a mix of the Cu_2_O of cubic and the CuO of the monoclinic structure phases with a polycrystalline nature in addition to Cu but existing in a very small quantity. It reveals two peaks related to Cu_2_O and located at 2θ = 36.4° and 42.4° with a reflection of (111) and (200) and two peaks related to CuO at 2θ = 60.4° and 65.4° due to the XRD from the (113) and (022) planes, respectively [20,21]. There are two peaks correlated to the FTO at 37.6° and 51.4° [22]. No other peaks relating to K or K_2_O were detected. This shows that the potassium ions added were well incorporated into the crystal lattice [23]. The formation Cu_2_O phase is predominant in the prepared samples and the doping supports the growth of the Cu_2_O along (111) in a vertical direction [24].

The average crystal size (D) of the fabricated thin films was determined by employing Debye–Scherrer’s equation [25] which is given by
D = 0.9λ/βCOSθ(1)
where D represents the crystal size, θ is the FWHM expressed in radians, λ is the X-ray wavelength and θ is the diffraction angle. The microstrain (ε) is estimated by the following relation [26]:ε = βcotθ/4(2)

The density of dislocation (δ) is approximated by the following equation [26]:δ = 1/D^2^(3)

The determined values for the microstructural properties are summarized in Table 1. By increasing the doped concentration from 0 to 0.05 M, the crystallite size decreased from 55 to 25.2 nm which may be attributed to K^+^ = 1.38 Å [27] which has a larger ionic radii than Cu^+^ in Cu_2_O (0.77 Å) [28] and Cu^+2^ in CuO (0.73 Å) [29], leading to the distortion of the local structure around the dopant site, and consequently a smaller lattice constant is produced [30]. With an increase in the dopant concentration, the crystallite size rises to about 29.1 nm, where K ions are located in an interstitial position near the Cu sites in the crystal lattice of Cu_2_O that reduces the defects [28].

### 3.2. Raman Analysis

Raman spectroscopy is utilized to investigate dopant incorporation, defects and structural disorders existent in the samples [23]. Figure 2 represents the Raman spectra of the measured samples in the range between 100 and 800 cm^−1^ which contain a mixed phase (Cu_2_O and CuO). From the analysis, there are three phonon modes at 148, 213 and 613 cm^−1^ related to Cu_2_O and corresponding to the Г-15, 2Г-12 and Γ15-(2) modes, respectively [31]. In addition, there are two peaks around 275 cm^−1^ and 323 cm^−1^ which are assigned to the A1g and Bg modes in CuO, respectively [32]. The two peaks come from the vibration of the oxygen atoms in CuO. When a pure sample is doped with potassium, no impurity peaks were identified which will support the XRD results. It is noted that the Raman peaks moved to a lower wavenumber at higher dopant concentrations. The Raman shift is due to a short-range order, and the interstitial of K+ near Cu sites will affect the long-range disorder and short-range order as a result of the lattice defect result from the difference in the charge between K^+^ ions, Cu+ and Cu^2+^ [33].

### 3.3. Surface Morphology Evaluation

Figure 3 represents the images of the SEM for pure and K−doped n−Cu_2_O thin films fabricated at different concentrations on the FTO glass substrate. It is clear that there are spherical grains on the sample that have been agglomerated. This agglomeration decreases with the increase in doping up to 0.05 M and then increases when the doping further increases. Such behavior could be explained by the vertical crystal growth which is slower than the lateral growth up to 0.05 M and then the growth rate inverts as the K ions increase, causing large grains.

### 3.4. Photoluminescence (PL) Evaluation

PL emission is caused by the recombination of free carriers and hence the PL spectra can be utilized to examine the efficiency of trapping charges. Figure 4 displays the PL spectra of all the samples at room temperature, with an excitation wavelength of 320 nm, and the measurements were performed in the range between 300 and 1000 nm. The PL spectrum displays two peaks: the first peak around 520 nm is related to the interband transitions in the Cu ions [34], and the second at 690 nm is related to the defects, including the copper interstitials in the CuO (oxygen vacancies) [35]. For the emission band correlated to interband transitions, the intensity increases up to 0.05 M. This behavior could be demonstrated by the inclusion of K ions in the Cu positions [34]. The strength of this peak declines after 0.05 M because of the inclusion of K ions in the interstitial sites [34]. These findings are in good agreement with the XRD patterns. With the emission band related to oxygen vacancies, the emission intensity increases up to 0.05 K ions and then decreases. The increase is due to doping with K ions which disturbs the crystal lattice. Therefore, the Cu—O bond is broken, and many oxygen vacancies are generated [34]. As the concentration of K ions increases, more nonradiative oxygen vacancy centers also rise and more photoexcited electrons get trapped in those oxygen vacancies, reducing the ability for recombination with the holes [34]. Such a result indicates the suitability of doped Cu_2_O thin films for photovoltaic applications [36].

### 3.5. Optical Analysis

Figure 5 displays a strong absorption edge for all the fabricated samples under visible light in the range between 425 nm and 460 nm which corresponds with the band gap of Cu_2_O [37].

The absorption spectrum was used to estimate the band gap (Eg) by employing Tauc’s plot relationship, which is given by [38,39,40]
(αhυ) = A(hυ − Eg)^n^(4)
where A is a constant, hν represents the photon energy, n is an exponent and related to the electronic transition which take the values 2 for indirect and ½ for direct transitions and α is the absorption coefficient which can be evaluated by
α = 2.303(A/T)(5)
where A is the absorbance and T represents the electrodeposited thin film thickness. Figure 6 shows the plots of (αhν)2 vs. (hν) and the estimated values of the Eg for the doped Cu_2_O thin films, which are listed in Table 1, that have values between 2.21 and 2.39 eV. The results agree well with other previous results for Cu_2_O thin film [41,42]. It is obvious that the estimated Eg gradually rises from 2.33 eV to 2.39 eV with a rising dopant concentration up to 0.05 M. This may be due to the taking of the K+ ions on the Cu sites which result in some distortion around the dopant site, leading to a decrease in the crystallite size, and quantum confinement was obtained [43]. The value of Eg then decreases with an increase in the dopant concentration to 2.21 at 0.1 M, as shown in Figure 7. This may be due to the growth of the band tail states that arise from increasing the doping content, producing a reduction in the value of the band gap [44].

### 3.6. Photocurrent and Photoelectrochemical Measurements

A photocurrent for pure and doped n−Cu_2_O thin films with different K ion concentrations was recorded, utilizing chronoamperometry to examine the photoelectrochemical properties of the electrodeposited thin films [45], as demonstrated in Figure 8. The pure and doped Cu_2_O thin films were employed as the working electrode for the three-electrode system Bio−LogicSb−50 potentiostat, and the measurements were completed on the aqueous solution of 0.5 M Na_2_SO_3_ at 0 V (vs. Ag/AgCl) under the light of a 200 W Tungsten/Halogen lamp. It is clear that photogenerated currents possess positive values, suggesting that Cu_2_O samples are n−type semiconductors [46]. When samples are subjected to light, the photocurrent increases rapidly and maintains this trend during illumination, but as soon as the light is turned off, the intensity of the current result from the light suddenly decreases. The density of the photocurrent has the highest value of 0.047 mA/cm^2^ for doped n−Cu_2_O thin films at 0.1 M K ions which indicates an enhancement in the transfer of charge carriers and a lower recombination rate. Such a significant enhancement in the photocurrent could result from the formation of dopant levels in the forbidden gap that increase the photoexcited electrons from the valence band [45]. So, the Eg value is decreased, leading to the creation of more electron–hole pairs and therefore the photogenerated current is increased. A 0.05 M sample has the lowest photocurrent, and this is because of its smaller particle size and higher band-gap value which effects the increase in the recombination charge carriers, as illustrated in the PL measurements. This is in agreement with the XRD, UV and PL results. Our findings confirm that the K−doped Cu_2_O thin films are excellent candidates for solar cell and biosensor applications.

### 3.7. Mott–Schottky and Electrochemical Impedance Spectroscopy (EIS) Measurements

The Mott–Schottky measurements were carried out as shown in Figure 9 to investigate the carrier density and the conductivity’s nature of the fabricated thin films (p-type or n-type) [47,48]. According to the Mott–Schottky equation, the semiconductor’s capacitance (C), the carrier concentration (Nd) and the other parameters such as the dielectric constant (ε for Cu_2_O = 7.6), vacuum permittivity (ε_0_), charge of electron (e), Boltzmann constant (k_B_), active surface area of the photoelectrode (A), temperature (T), applied potential (v) and flat band potential (Vfb) are related by the following equation [48]:(6)$$\frac{1}{C^{2}}=\frac{2}{-{\varepsilon \varepsilon }_{0}{\mathrm{eA}}^{2}\;N_{d}\;}\left(V\;-{\;V}_{\mathrm{fb}}-\frac{\;\mathrm{kT}}{e\;}\right)$$

The interfacial capacitances, at the interface of the semiconductor/electrolyte, are obtained from the electrochemical impedance measured at the potential ranging from −1.2 to −0.05 V with an AC perturbation frequency of 10 kHz and with an amplitude of 5 mV [49]. The Vfb was determined experimentally by extrapolating the linear section of the Mott–Schottky plots and placing the intercept on the *x*-axis [50]. The value of the flat band potential was found to decline from −0.8 to −0.71 V vs. with the increase in the doping concentration from 0 to 0.05 M for K ions, but with further growth in the doping concentration, the flat band potential increased up to −0.87 V vs. Flat band potential that is more negative has the ability to assist the charge separation at the interface of the semiconductor/electrolyte [51]. It can be noted that the maximum flat band potential for 0.1 M of Cu_2_O doped with K ions was observed, which is consistent with the maximum photocurrent density displayed by this sample [52]. The carrier concentration was calculated utilizing the following equation [50].
(7)$$\;N_{D}=\frac{2}{-{\varepsilon \varepsilon }_{0}{\mathrm{eA}}^{2}\;S\;}$$
where S is the plot’s slope of Mott–Schottky and can be described as [50]
(8)$$S={\left[\frac{d}{dv}\left(\frac{1}{c^{2}}\right)\right]}^{-1}$$

The Mott–Schottky plots for the pure and K−doped Cu_2_O exhibited a positive slope, signifying that all the Cu_2_O thin films were n-type semiconductors. The donor density raised from 1.3 × 10^17^ to 3.2 × 10^17^ cm^−3^ with the increase in the doping concentration, indicating a higher density of the vacancies in the doped samples as presented in Table 2 [53]. This may be due to two reasons: The first could be caused by the substitution of Cu^+2^ in Cu_2_O by K^+^ and producing oxygen vacancies to maintain the charge neutrality [36]. The second reason may be due to potassium acting as the donor level, and its incorporation in the crystal increases the donor density [52,54].

The electrochemical impedance spectroscopy (EIS) measurements were utilized to identify the interfacial charge-transfer behavior for all the samples in an electrolyte [49]. These measurements were made in 0.5 M Na_2_SO_4_ at a perturbation potential of 0.5 V at the frequency range of 10^4^ Hz. Figure 10 exhibits the Nyquist plot (Z imaginary vs. Z real) for the Cu_2_O samples, which was fitted by ZSimpWin software with the equivalent electrical circuit model of R(Q(R(Q(RW)))), including the solution resistance (R1), charge-transfer resistance (R2), adsorption resistance (R3), constant phase element (Q) and Warburg’s impedance (W) [55]. The Nyquist plots include semicircles, and the diameter equals the resistance to charge transfer (R_CT_) across the electrode/electrolyte interface. The greater the arc diameter, the greater the charge-transfer resistance [56]. K ions-doped Cu_2_O thin films produced at 0.1 M are found to have a diameter that is significantly smaller than the diameters of the other semicircles, indicating a faster charge transfer and a lower rate of charge recombination, which is talented for p−Cu_2_O in homojunctions solar cells [36].

## 4. Conclusions

In conclusion, pure and K−doped n−Cu_2_O thin films were successfully electrodeposited by the electrochemical technique on FTO substrates. Their microstructural, morphological, optical and photoelectrochemical properties have been studied. The electrodeposited samples were grown in a mixed phase with cubic Cu_2_O and monoclinic CuO, but the Cu_2_O phase is predominant in the prepared samples with a preferable (111) orientation and the crystallite size is in the range between 55 and 25.2 nm. The Raman analysis confirmed the presence of functional groups relating to Cu_2_O and CuO in the thin films. The SEM results revealed spherical grains on the thin films that were agglomerated, and this agglomeration could be affected by K doping. The PL spectra demonstrated two peaks at 520 and 690 nm relating to interband transitions in Cu_2_O and oxygen vacancies in CuO, respectively. Moreover, the PL intensity could be quenched at higher dopant concentrations which shows that the dopant K limits the e−/h+ pairs recombination by trapped electrons and holes. The optical absorption of the prepared samples showed a redshift in Eg. The photocurrent measurements displayed that the produced thin films are n-type semiconductors. Moreover, the photocurrent was enhanced by a growing doping concentration where 0.1 M of K ions has the highest value. The Mott– Schottky measurements revealed that the flat band potential and donor density vary with the doping concentration from −0.87 to −0.71 V and 1.3 × 10^17^ to 3.2 × 10^17^ cm^−3^, respectively. EIS shows that the lowest resistivity to charge transfer (Rct) was attained at a 0.1 M concentration of K ions. From the outcomes, it appears that K−doped n−Cu_2_O thin films are an excellent candidate for a solar cell and biosensor application.

## Figures and Tables

**Figure 1 nanomaterials-13-01272-f001:**
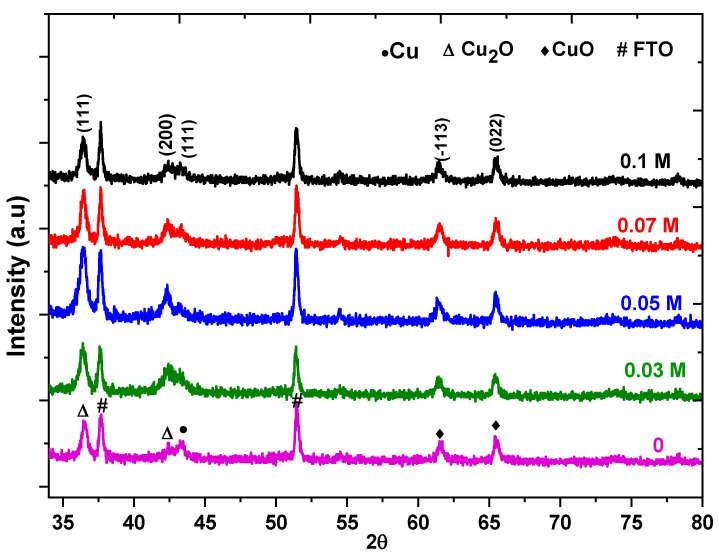
XRD patterns of n−Cu_2_O thin films doped with different K ions concentrations.

**Figure 2 nanomaterials-13-01272-f002:**
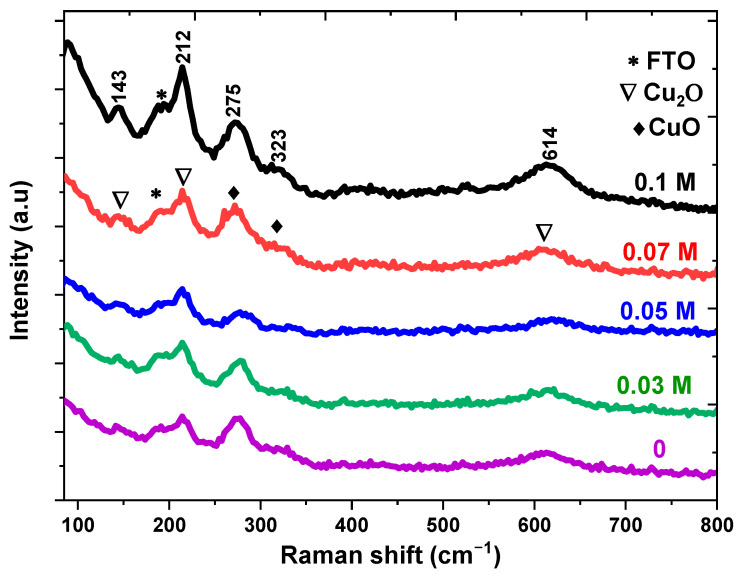
Raman spectra of K−doped n−Cu_2_O thin films with different concentrations.

**Figure 3 nanomaterials-13-01272-f003:**
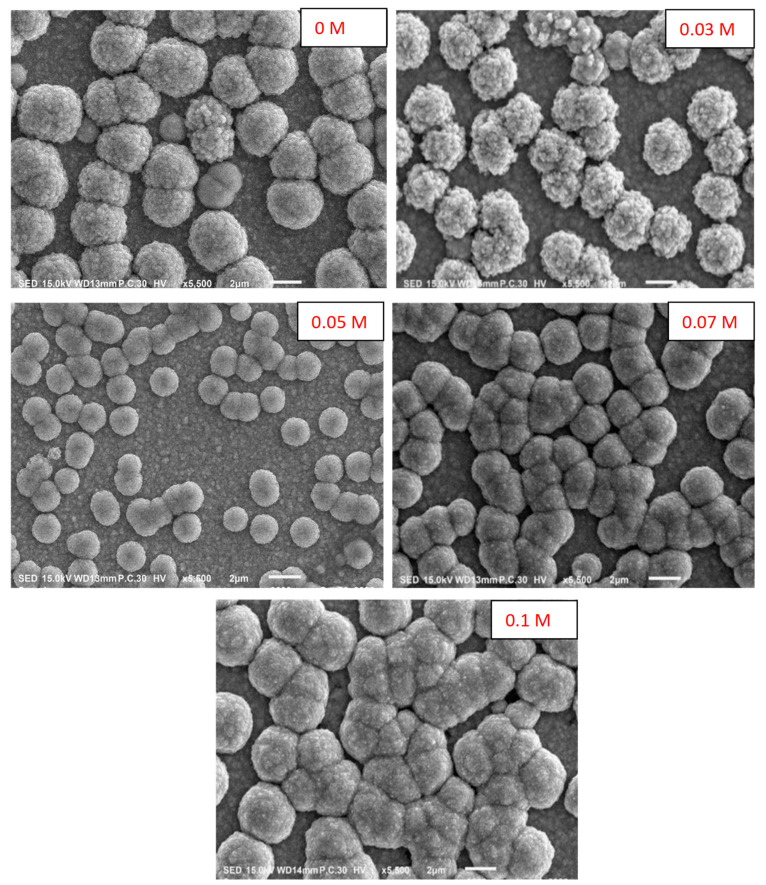
SEM photographs of K−doped n−Cu_2_O thin films electrodeposited as n−type at different concentrations.

**Figure 4 nanomaterials-13-01272-f004:**
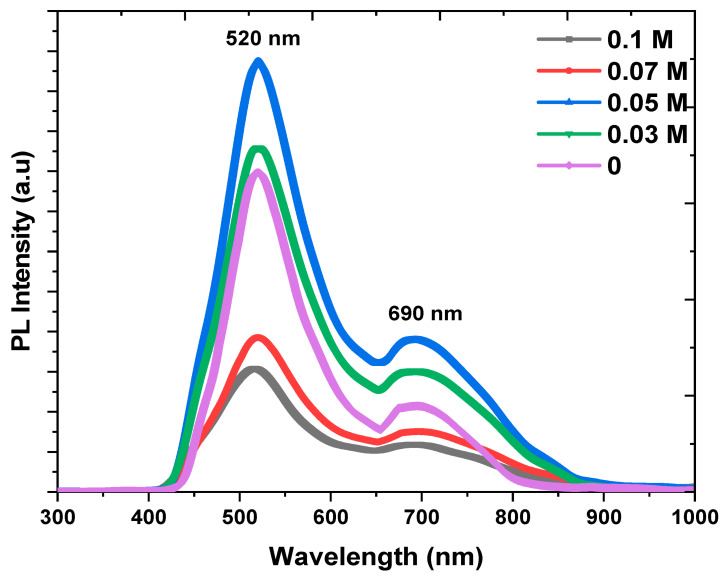
PL spectra of K−doped n−Cu_2_O thin films with different concentrations.

**Figure 5 nanomaterials-13-01272-f005:**
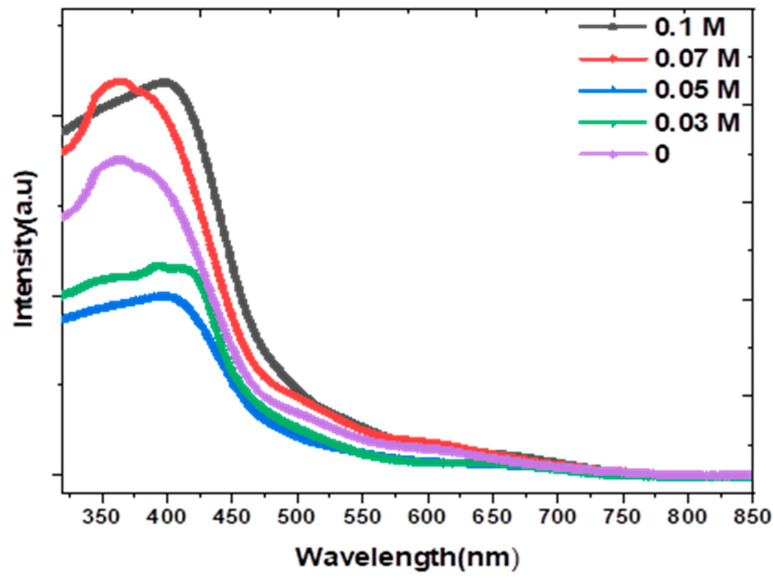
UV−Vis absorption spectra of K−doped n−Cu_2_O thin films with different concentrations.

**Figure 6 nanomaterials-13-01272-f006:**
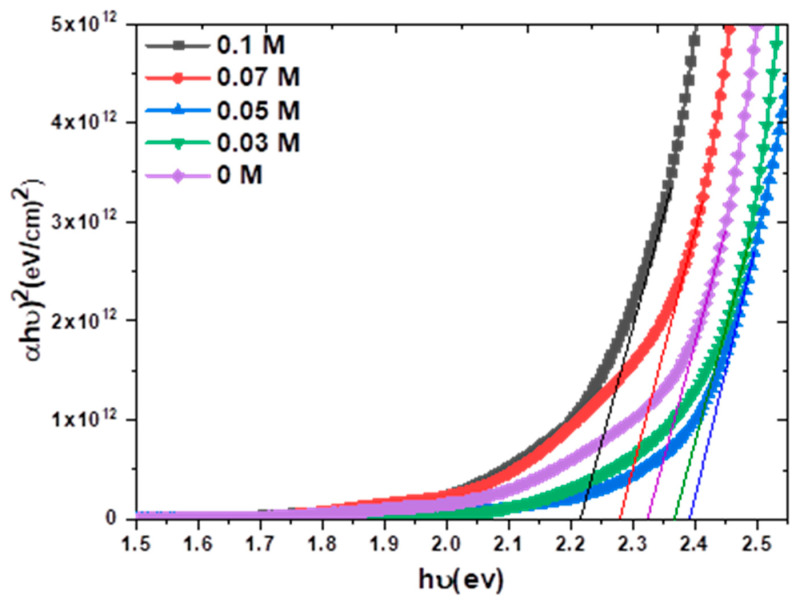
Band gap of K−doped Cu_2_O thin films electrodeposited as n−type at different concentrations.

**Figure 7 nanomaterials-13-01272-f007:**
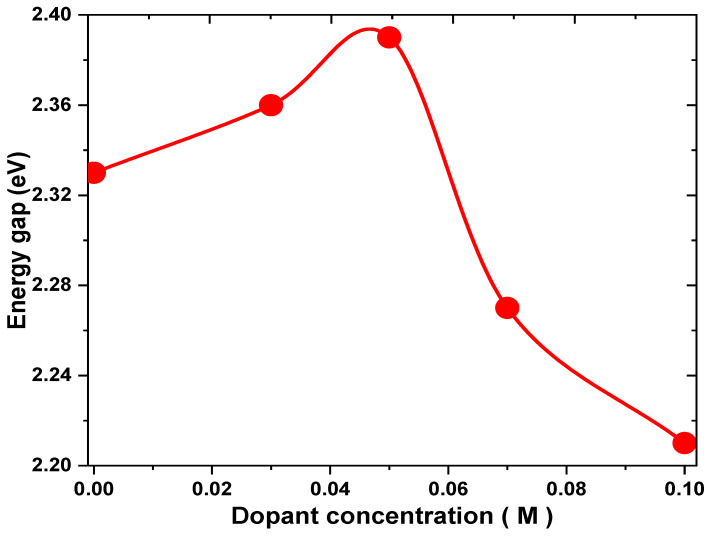
Variation in Eg for Cu_2_O thin films at different concentrations of K ions.

**Figure 8 nanomaterials-13-01272-f008:**
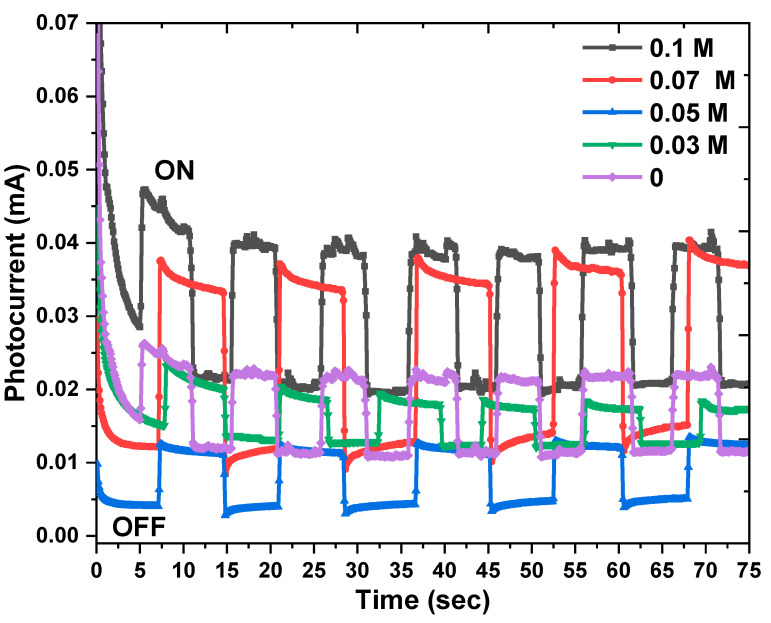
Photocurrent measurements of K−doped Cu_2_O thin films electrodeposited as n-type at different dopant concentrations.

**Figure 9 nanomaterials-13-01272-f009:**
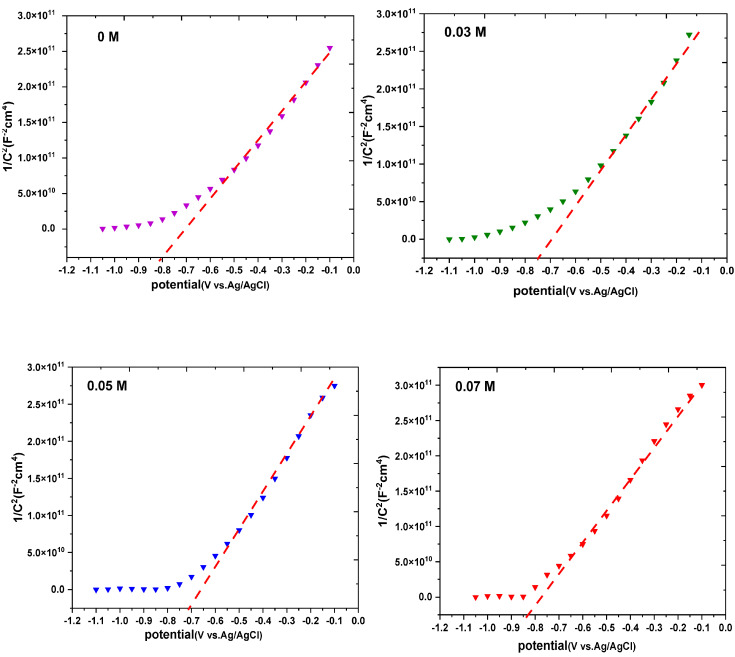
Mott−Schottky measurements for K−doped Cu_2_O with different concentrations (dashed lines present the linear part of plots thar intercept on the *x*-axis).

**Figure 10 nanomaterials-13-01272-f010:**
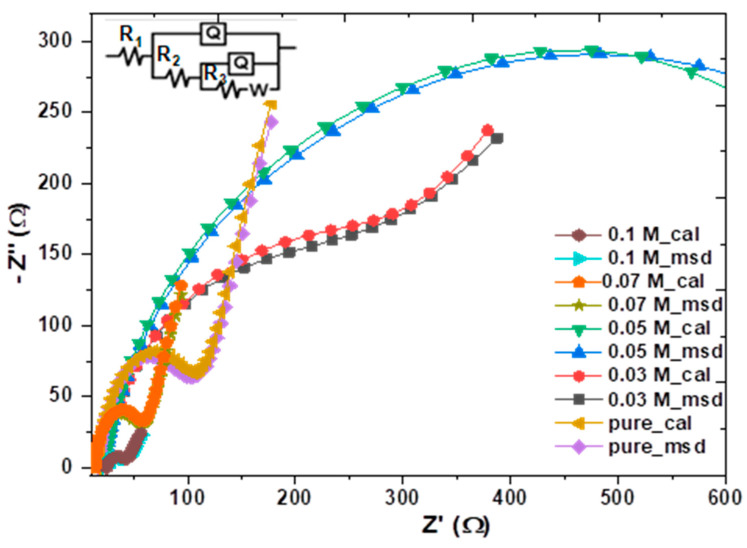
Nyquist plots (Z real versus Z imaginary) for undoped and K−doped n−Cu_2_O thin films measured in 0.5 M Na_2_SO_4_.

**Table 1 nanomaterials-13-01272-t001:** Structural parameter of fabricated thin films at different concentrations.

Doped Concentration	Crystallite Size Avg (D) (nm)	Dislocation Density (δ) × 10^14^	Microstrain (ε) ×10^−3^ lines^−2^ m^4^	Energy Gap (eV)
0	55	3.3	8.4	2.33
0.03	31.2	10.2	17	2.35
0.05	25.2	15.7	19.7	2.37
0.07	27.8	12.9	17.8	2.26
0.1	29.1	11.8	16.3	2.21

**Table 2 nanomaterials-13-01272-t002:** Influence of doping with K ions on carrier densities and flat band potentials for Cu_2_O thin films.

Samples	V_fb_ (V vs. Ag/AgCl)	Conductivity Type	Dopant’s Density (cm^− 3^)
Undoped Cu_2_O	−0.8	n-type	1.3 × 10^17^
0.03 K-Cu_2_O	−0.75	n-type	1.7 × 10^17^
0.05 K-Cu_2_O	−0.71	n-type	2.1 × 10^17^
0.07 K-Cu_2_O	−0.83	n-type	2.6 × 10^17^
0.1 K-Cu_2_O	−0.87	n-type	3.2 × 10^17^

## Data Availability

Data available on request.

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
