# Peer review of "Enhancement of Structural, Optical and Photoelectrochemical Properties of n−Cu2O Thin Films with K Ions Doping toward Biosensor and Solar Cell Applications"

_nanomaterials, 2023, doi:10.3390/nano13071272_

Round 1

Reviewer 1 Report

The manuscript “Enhancement of structural, optical and photoelectrochemical properties of n-Cu2O thin films with K ions doping towards biosensor and solar cell applications” deals with the fabrication and characterization of Cu2O thin films, with and without K ions doping. The thin films are obtained by electrochemical deposition and characterized by a variety of characterization techniques. The focus is on the structural properties and their optical properties. The main applications envisaged are solar cells and biosensors, the optical and electrical response to stimulation with light being analyzed.

The manuscript is well written, the results are interesting and presented in a logical succession, easy to follow. Some minor clarifications and corrections are needed before publication:

A careful reading of the manuscript is needed to correct the English language. Some examples will be given in the following, section by section.

In the abstract,

Please be consistent in the enumeration: spectrometry, spectroscopy.., photocurrent measurements.

Lines 20 -22, please rephrase.

Line 29 please replace “doping” with dopped”

Introduction

Line 45, “hydrothermal method” is repeated

Line 47, please rephrase “is well done”

Materials and Methods: Line 70, please replace “doing” with “doping”

Results and discussions:

In the Results and Discussion section please use the same colors for the graphical representation, keep the same codification for the 5 samples that are analyzed. It is now confusing because each graphic has a different color codification.

Line 106, please rephrase

Line 109, please rephrase “is larger” with “has a larger”

Lines 111-115, please rephrase, the phrase is too long and difficult to follow.

In table 1, please check the name on the first line. Is it PH value or K ions concentration?

In figure 3 , SEM images are presented, with spherical grains on the surface. Please specify if only the grains represent the “thin film” or there is a continuous layer with spherical grains on top of it. If it is the latter, please evaluate the thickness of the continuous layer in comparison with the spherical particles.

Line 168, and all the other occurrences in the text: please replace “until 0.05 M” with “up to 0.05 M” or something similar.

Line 170 , please replace “have a well agreement” with “are in good agreement”

Line 176-177, please clarify or rephrase. What does “probable” mean?

Line 248-249, please rephrase, avoid using “represent as traps”

Figure 8, the photocurrent in dark conditions for 0.05 M is smaller compared to the others. Is there a reason for this? 0.05 M seems to be a critical point, from the other characterizations.

Author Response

List of changes and answers to the reviewer’s comments

Dear editors and reviewers,

First of all, we would like to thank editors and reviewers for their valuable effort in reviewing our manuscript and constructive remarks throughout the reports.

We have modified our manuscript accordingly to the reviewer’s comments as listed below point by point:

Response to Reviewer 1 comments:

The manuscript “Enhancement of structural, optical and photoelectrochemical properties of n-Cu2O thin films with K ions doping towards biosensor and solar cell applications” deals with the fabrication and characterization of Cu2O thin films, with and without K ions doping. The thin films are obtained by electrochemical deposition and characterized by a variety of characterization techniques. The focus is on the structural properties and their optical properties. The main applications envisaged are solar cells and biosensors, the optical and electrical response to stimulation with light being analyzed.

The manuscript is well written, the results are interesting and presented in a logical succession, easy to follow. Some minor clarifications and corrections are needed before publication:

A careful reading of the manuscript is needed to correct the English language. Some examples will be given in the following, section by section.

In the abstract,

Please be consistent in the enumeration: spectrometry, spectroscopy.., photocurrent measurements.

The sentence change to be

Doping of n-Cu2O thin films with K ions was well identified using XRD, Raman, SEM, EDX, UV-vis, PL, photocurrent, Mott– Schottky, and EIS measurements

Lines 20 -22, please rephrase.

Moreover, the samples crystallinity and morphology change by doping concentrations which confirmed by SEM.

Line 29 please replace “doping” with dopped”

The word is changed.

Introduction

Line 45, “hydrothermal method” is repeated

The repeated words are deleted.

Line 47, please rephrase “is well done”

 The sentence is changed to

Cu2O is a promising candidate for photovoltaic energy conversion.

Materials and Methods: Line 70, please replace “doing” with “doping”

 The word is changed.

Results and discussions:

In the Results and Discussion section please use the same colors for the graphical representation, keep the same codification for the 5 samples that are analyzed. It is now confusing because each graphic has a different color codification.

The same colors are used for all the graphical representation.

Line 106, please rephrase

The sentence is rephrased as:

With increasing dopped concentration from 0 to 0.05 M, the crystallite size decreased from 55 to 25.2 nm which may be attributed K+= 1.38 Å)  [27] which has a larger ionic radii than Cu+ in Cu2O (0.77 Å)

Line 109, please rephrase “is larger” with “has a larger”

 The words are changed.

Lines 111-115, please rephrase, the phrase is too long and difficult to follow.

The sentence is rephrased as:

With increase of dopant concentration, the crystallite size rises to about 29.1 nm where K ions are located in interstitial position near Cu sites in the crystal lattice of Cu2O that re-duce the defects [28].

In table 1, please check the name on the first line. Is it PH value or K ions concentration?

Thanks for such comments, it is in fact dopped concentration, and it is changed.

In figure 3 , SEM images are presented, with spherical grains on the surface. Please specify if only the grains represent the “thin film” or there is a continuous layer with spherical grains on top of it. If it is the latter, please evaluate the thickness of the continuous layer in comparison with the spherical particles.

In SEM, the grain represent the “thin film” and this also cleared from XRD measurements where FTO substrate peak is appeared.

Line 168, and all the other occurrences in the text: please replace “until 0.05 M” with “up to 0.05 M” or something similar.

The word is changed.

Line 170 , please replace “have a well agreement” with “are in good agreement”

The words are changed.

Line 176-177, please clarify or rephrase. What does “probable” mean?

The sentence is rephrased to be

Such result indicates the suitable of dopped Cu2O thin films for photovoltaic applications.

Line 248-249, please rephrase, avoid using “represent as traps”

The word are rephrased to be

that increase photoexcited electrons from the valence band

Figure 8, the photocurrent in dark conditions for 0.05 M is smaller compared to the others. Is there a reason for this? 0.05 M seems to be a critical point, from the other characterizations.

0.05 M sample has the lowest photocurrent and this because of its smaller particle size and higher band gap value which effects on increasing of recombination charge carries as illustrate from PL measurements.

Abdelfatah et al. presented n-type Cu2O films on FTO substrates using electrodeposition method and characterized the films with XRD, Raman, SEM, EDX, UV-vis spectrophotometer, PL spectroscopy, and photocurrent measurement. The work lacks novelty and the data shown in the manuscript is not solid enough for publication. I suggest to reconsider after major revision for the current manuscript. Please see below for my comments:

  1. The introduction should justify why the Cu2O films on FTO substrates can be of interest to the research community. What's the new in their studies when comparing to those cited references?

In the revised manuscript, the following sentences are added to clear this point:

In this work, n-Cu2O was potentiostatically electrodeposited on FTO conductive substrates where many optoelectronic devices such solar cell need front and back contacts to collect the charge carriers and therefore FTO could be used as electrode for devices.

  1. The authors claimed the proposed material system can be an excellent candidate for biosensor and solar cell applications, while I cannot see any demonstrations but only standard characterizations.

Thanks a lot for such comment, in fact we started only to fabricate doped Cu2O films and tested photocurrent measurements as first step for biosensor and solar cell applications. Nevertheless, we make new measurements such as Mott-Schottky and Electrochemical impedance spectroscopy (EIS) measurements to prove that as following:

Mott-Schottky and Electrochemical impedance spectroscopy (EIS) measurements:

Mott-Schottky measurements were carried out in as shown in Figure 9 to investigate the carrier density and the conductivity's nature of the fabricated thin films (p-type or n-type) [48].  According to the Mott–Schottky equation, the semiconductor’s capacitance (C), the carrier concentration (Nd) and the other parameters such as dielectric constant (ε for Cu2O= 7.6), vacuum permittivity (ε0), the charge of electron (e) Boltzmann constant (kB) the active surface area of the photoelectrode (A), temperature (T), the applied potential (v) and the flat-band potential (Vfb) are related by the following equation [48]:

                       (6)    

The interfacial capacitances, at the interface of the semiconductor/electrolyte, are obtained from the electrochemical impedance measured at potential ranged from -1.2 to -0.05 V with an AC perturbation frequency of 10 kHz and with the amplitude of 5 mV [49]. Vfb was determined experimentally by extrapolating the linear section of the Mott-Schottky plots and placing the intercept on the x-axis  [50]. The value of flatband potential was found to decline from -0.8 to -0.71 V vs with the increase of doping concentration from 0 to 0.05 M for K ions but further growth in the doping concentration, the flatband potential increased up to -0.87 V vs. Flatband potential that is more negative has the ability to assist the charge separation at the interface of the semiconductor/electrolyte [51]. It can be noted that the maximum flatband potential for 0.1M of Cu2O doped with K ions was observed, which is consistent with the maximum photocurrent density displayed by this sample [52]. The carrier concentration was calculated utilizing the following equation [50].

                                                            (7)    

where S is the plot’s slope of  Mott-Schottky and can be described as [50] :

                                                            (8)

Mott-Schottky plots for pure and K doped Cu2O exhibited a positive slope, signifying that all Cu2O thinfilms were n-type semiconductors. The donor density raised from 1.3 × 1017 to 3.2 × 1017 cm-3 with the increase in doping concentration, indicating a higher density of the vacancies in doped samples [53]. This may be due to two reasons: the first could be caused by substitution of Cu+2 in Cu2O by K+ and producing oxygen vacancies to maintain the charge neutrality [36]. The second reason may be due to that potassium acts as donor level and its incorporation in the crystal increases donor density [52, 54].

Table 2.  Influence of doping with K ions on carrier densities and flat band potentials for n-Cu2O thin films.

Samples

Vfb

(V vs. Ag/AgCl)

Conductivity type

Dopant’s density

(cm− 3)

Un doped Cu2O

-0.8

n-type

1.3×1017

0.03 K-Cu2O

-0.75

n-type

1.7×1017

0.05 K-Cu2O

-0.71

n-type

2.1×1017

0.07 K-Cu2O

-0.83

n-type

2.6×1017

0.1 K-Cu2O

-0.87

n-type

3.2×1017

Figure 9. Mott–Schottky measurements for K doped Cu2O with different concentrations.

Electrochemical impedance spectroscopy (EIS) measurements were utilized to identify the interfacial charge-transfer behavior for all samples in an electrolyte [49]. These measurements were made in 0.5 M Na2SO4 at a perturbation potential of 0.5 V  at the frequency range of 104 Hz. Figure 10 exhibits the Nyquist plot (Z imaginary vs. Z real) for Cu2O samples, which was fitted by a ZSimpWin software with the equivalent electrical circuit model of R(Q(R(Q(RW)))), including the solution-resistance (R1), charge-transfer resistance (R2), the adsorption resistance (R3), constant phase element (Q), Warburg’s impedance (W) [55]. The Nyquist plots include semicircles and its diameter equals the resistance to charge transfer (RCT) across the electrode/electrolyte interface. The more the arc diameter, the more the charge transfer resistance [56]. K ions doped Cu2O thin films and produced at 0.1 M are found to have a diameter that is significantly smaller than the diameters of the other semicircles, indicating faster charge transfer and lower rate of charge recombination, which is talented for p-Cu2O in homojunctions solar cells [36].

Figure 10. Nyquist plots (Z real versus Z imaginary) for undoped and K-doped n-Cu2O thin films measured in 0.5 M Na2SO4.

  1. In Figure 1, the symbol for CuO and FTO is quite similar for me (upper right corner of the figure), please use symbols that are easy to distinguish.

In the revised manuscript, the symbol for FTO was changed to #.

  1. According to Table 1 and Fig. 7, the doping level cannot quite efficiently modulate the energy gap. It makes me doubt whether doping can be an efficient way for improving the energy conversion efficiency of the system.

In fact, the doping level effect in both band gap and carrier concentration and consequently the energy conversion efficiency of the system. So, we measure the carrier concentration as following:

Mott-Schottky measurements were carried out in as shown in Figure 9 to investigate the carrier density and the conductivity's nature of the fabricated thin films (p-type or n-type) [48].  According to the Mott–Schottky equation, the semiconductor’s capacitance (C), the carrier concentration (Nd) and the other parameters such as dielectric constant (ε for Cu2O= 7.6), vacuum permittivity (ε0), the charge of electron (e) Boltzmann constant (kB) the active surface area of the photoelectrode (A), temperature (T), the applied potential (v) and the flat-band potential (Vfb) are related by the following equation [48]:

                       (6)    

The interfacial capacitances, at the interface of the semiconductor/electrolyte, are obtained from the electrochemical impedance measured at potential ranged from -1.2 to -0.05 V with an AC perturbation frequency of 10 kHz and with the amplitude of 5 mV [49]. Vfb was determined experimentally by extrapolating the linear section of the Mott-Schottky plots and placing the intercept on the x-axis  [50]. The value of flatband potential was found to decline from -0.8 to -0.71 V vs with the increase of doping concentration from 0 to 0.05 M for K ions but further growth in the doping concentration, the flatband potential increased up to -0.87 V vs. Flatband potential that is more negative has the ability to assist the charge separation at the interface of the semiconductor/electrolyte [51]. It can be noted that the maximum flatband potential for 0.1M of Cu2O doped with K ions was observed, which is consistent with the maximum photocurrent density displayed by this sample [52]. The carrier concentration was calculated utilizing the following equation [50].

                                                            (7)    

where S is the plot’s slope of  Mott-Schottky and can be described as [50] :

                                                            (8)

Mott-Schottky plots for pure and K doped Cu2O exhibited a positive slope, signifying that all Cu2O thinfilms were n-type semiconductors. The donor density raised from 1.3 × 1017 to 3.2 × 1017 cm-3 with the increase in doping concentration, indicating a higher density of the vacancies in doped samples [53]. This may be due to two reasons: the first could be caused by substitution of Cu+2 in Cu2O by K+ and producing oxygen vacancies to maintain the charge neutrality [36]. The second reason may be due to that potassium acts as donor level and its incorporation in the crystal increases donor density [52, 54].

Table 2.  Influence of doping with K ions on carrier densities and flat band potentials for n-Cu2O thin films.

Samples

Vfb

(V vs. Ag/AgCl)

Conductivity type

Dopant’s density

(cm− 3)

Un doped Cu2O

-0.8

n-type

1.3×1017

0.03 K-Cu2O

-0.75

n-type

1.7×1017

0.05 K-Cu2O

-0.71

n-type

2.1×1017

0.07 K-Cu2O

-0.83

n-type

2.6×1017

0.1 K-Cu2O

-0.87

n-type

3.2×1017

Figure 9. Mott–Schottky measurements for K doped Cu2O with different concentrations.

  1. The photoelectrochemical measurements are too simple to demonstrate the corresponding properties of the system. The authors may need to add a full set of data to characterize the photoelectrochemical properties.

Thanks for such comment, and for that we add Electrochemical impedance spectroscopy (EIS) measurements as following:

Electrochemical impedance spectroscopy (EIS) measurements were utilized to identify the interfacial charge-transfer behavior for all samples in an electrolyte [49]. These measurements were made in 0.5 M Na2SO4 at a perturbation potential of 0.5 V  at the frequency range of 104 Hz. Figure 10 exhibits the Nyquist plot (Z imaginary vs. Z real) for Cu2O samples, which was fitted by a ZSimpWin software with the equivalent electrical circuit model of R(Q(R(Q(RW)))), including the solution-resistance (R1), charge-transfer resistance (R2), the adsorption resistance (R3), constant phase element (Q), Warburg’s impedance (W) [55]. The Nyquist plots include semicircles and its diameter equals the resistance to charge transfer (RCT) across the electrode/electrolyte interface. The more the arc diameter, the more the charge transfer resistance [56]. K ions doped Cu2O thin films and produced at 0.1 M are found to have a diameter that is significantly smaller than the diameters of the other semicircles, indicating faster charge transfer and lower rate of charge recombination, which is talented for p-Cu2O in homojunctions solar cells [36].

Figure 10. Nyquist plots (Z real versus Z imaginary) for undoped and K-doped n-Cu2O thin films measured in 0.5 M Na2SO4.

Reviewer 2 Report

Abdelfatah et al. presented n-type Cu2O films on FTO substrates using electrodeposition method and characterized the films with XRD, Raman, SEM, EDX, UV-vis spectrophotometer, PL spectroscopy, and photocurrent measurement. The work lacks novelty and the data shown in the manuscript is not solid enough for publication. I suggest to reconsider after major revision for the current manuscript. Please see below for my comments:

1. The introduction should justify why the Cu2O films on FTO substrates can be of interest to the research community. What's the new in their studies when comparing to those cited references?

2. The authors claimed the proposed material system can be an excellent candidate for biosensor and solar cell applications, while I cannot see any demonstrations but only standard characterizations. 

3. In Figure 1, the symbol for CuO and FTO is quite similar for me (upper right corner of the figure), please use symbols that are easy to distinguish.

4. According to Table 1 and Fig. 7, the doping level cannot quite efficiently modulate the energy gap. It makes me doubt whether doping can be an efficient way for improving the energy conversion efficiency of the system.

5. The photoelectrochemical measurements are too simple to demonstrate the corresponding properties of the system. The authors may need to add a full set of data to characterize the photoelectrochemical properties. 

Author Response

List of changes and answers to the reviewer’s comments

Dear editors and reviewers,

First of all, we would like to thank editors and reviewers for their valuable effort in reviewing our manuscript and constructive remarks throughout the reports.

We have modified our manuscript accordingly to the reviewer’s comments as listed below point by point:

Response to Reviewer 2 comments:

Abdelfatah et al. presented n-type Cu2O films on FTO substrates using electrodeposition method and characterized the films with XRD, Raman, SEM, EDX, UV-vis spectrophotometer, PL spectroscopy, and photocurrent measurement. The work lacks novelty and the data shown in the manuscript is not solid enough for publication. I suggest to reconsider after major revision for the current manuscript. Please see below for my comments:

  1. The introduction should justify why the Cu2O films on FTO substrates can be of interest to the research community. What's the new in their studies when comparing to those cited references?

In the revised manuscript, the following sentences are added to clear this point:

In this work, n-Cu2O was potentiostatically electrodeposited on FTO conductive substrates where many optoelectronic devices such solar cell need front and back contacts to collect the charge carriers and therefore FTO could be used as electrode for devices.

  1. The authors claimed the proposed material system can be an excellent candidate for biosensor and solar cell applications, while I cannot see any demonstrations but only standard characterizations.

Thanks a lot for such comment, in fact we started only to fabricate doped Cu2O films and tested photocurrent measurements as first step for biosensor and solar cell applications. Nevertheless, we make new measurements such as Mott-Schottky and Electrochemical impedance spectroscopy (EIS) measurements to prove that as following:

Mott-Schottky and Electrochemical impedance spectroscopy (EIS) measurements:

Mott-Schottky measurements were carried out in as shown in Figure 9 to investigate the carrier density and the conductivity's nature of the fabricated thin films (p-type or n-type) [48].  According to the Mott–Schottky equation, the semiconductor’s capacitance (C), the carrier concentration (Nd) and the other parameters such as dielectric constant (ε for Cu2O= 7.6), vacuum permittivity (ε0), the charge of electron (e) Boltzmann constant (kB) the active surface area of the photoelectrode (A), temperature (T), the applied potential (v) and the flat-band potential (Vfb) are related by the following equation [48]:

                       (6)    

The interfacial capacitances, at the interface of the semiconductor/electrolyte, are obtained from the electrochemical impedance measured at potential ranged from -1.2 to -0.05 V with an AC perturbation frequency of 10 kHz and with the amplitude of 5 mV [49]. Vfb was determined experimentally by extrapolating the linear section of the Mott-Schottky plots and placing the intercept on the x-axis  [50]. The value of flatband potential was found to decline from -0.8 to -0.71 V vs with the increase of doping concentration from 0 to 0.05 M for K ions but further growth in the doping concentration, the flatband potential increased up to -0.87 V vs. Flatband potential that is more negative has the ability to assist the charge separation at the interface of the semiconductor/electrolyte [51]. It can be noted that the maximum flatband potential for 0.1M of Cu2O doped with K ions was observed, which is consistent with the maximum photocurrent density displayed by this sample [52]. The carrier concentration was calculated utilizing the following equation [50].

                                                            (7)    

where S is the plot’s slope of  Mott-Schottky and can be described as [50] :

                                                            (8)

Mott-Schottky plots for pure and K doped Cu2O exhibited a positive slope, signifying that all Cu2O thinfilms were n-type semiconductors. The donor density raised from 1.3 × 1017 to 3.2 × 1017 cm-3 with the increase in doping concentration, indicating a higher density of the vacancies in doped samples [53]. This may be due to two reasons: the first could be caused by substitution of Cu+2 in Cu2O by K+ and producing oxygen vacancies to maintain the charge neutrality [36]. The second reason may be due to that potassium acts as donor level and its incorporation in the crystal increases donor density [52, 54].

Table 2.  Influence of doping with K ions on carrier densities and flat band potentials for n-Cu2O thin films.

Samples

Vfb

(V vs. Ag/AgCl)

Conductivity type

Dopant’s density

(cm− 3)

Un doped Cu2O

-0.8

n-type

1.3×1017

0.03 K-Cu2O

-0.75

n-type

1.7×1017

0.05 K-Cu2O

-0.71

n-type

2.1×1017

0.07 K-Cu2O

-0.83

n-type

2.6×1017

0.1 K-Cu2O

-0.87

n-type

3.2×1017

Figure 9. Mott–Schottky measurements for K doped Cu2O with different concentrations.

Electrochemical impedance spectroscopy (EIS) measurements were utilized to identify the interfacial charge-transfer behavior for all samples in an electrolyte [49]. These measurements were made in 0.5 M Na2SO4 at a perturbation potential of 0.5 V  at the frequency range of 104 Hz. Figure 10 exhibits the Nyquist plot (Z imaginary vs. Z real) for Cu2O samples, which was fitted by a ZSimpWin software with the equivalent electrical circuit model of R(Q(R(Q(RW)))), including the solution-resistance (R1), charge-transfer resistance (R2), the adsorption resistance (R3), constant phase element (Q), Warburg’s impedance (W) [55]. The Nyquist plots include semicircles and its diameter equals the resistance to charge transfer (RCT) across the electrode/electrolyte interface. The more the arc diameter, the more the charge transfer resistance [56]. K ions doped Cu2O thin films and produced at 0.1 M are found to have a diameter that is significantly smaller than the diameters of the other semicircles, indicating faster charge transfer and lower rate of charge recombination, which is talented for p-Cu2O in homojunctions solar cells [36].

Figure 10. Nyquist plots (Z real versus Z imaginary) for undoped and K-doped n-Cu2O thin films measured in 0.5 M Na2SO4.

  1. In Figure 1, the symbol for CuO and FTO is quite similar for me (upper right corner of the figure), please use symbols that are easy to distinguish.

In the revised manuscript, the symbol for FTO was changed to #.

  1. According to Table 1 and Fig. 7, the doping level cannot quite efficiently modulate the energy gap. It makes me doubt whether doping can be an efficient way for improving the energy conversion efficiency of the system.

In fact, the doping level effect in both band gap and carrier concentration and consequently the energy conversion efficiency of the system. So, we measure the carrier concentration as following:

Mott-Schottky measurements were carried out in as shown in Figure 9 to investigate the carrier density and the conductivity's nature of the fabricated thin films (p-type or n-type) [48].  According to the Mott–Schottky equation, the semiconductor’s capacitance (C), the carrier concentration (Nd) and the other parameters such as dielectric constant (ε for Cu2O= 7.6), vacuum permittivity (ε0), the charge of electron (e) Boltzmann constant (kB) the active surface area of the photoelectrode (A), temperature (T), the applied potential (v) and the flat-band potential (Vfb) are related by the following equation [48]:

                       (6)    

The interfacial capacitances, at the interface of the semiconductor/electrolyte, are obtained from the electrochemical impedance measured at potential ranged from -1.2 to -0.05 V with an AC perturbation frequency of 10 kHz and with the amplitude of 5 mV [49]. Vfb was determined experimentally by extrapolating the linear section of the Mott-Schottky plots and placing the intercept on the x-axis  [50]. The value of flatband potential was found to decline from -0.8 to -0.71 V vs with the increase of doping concentration from 0 to 0.05 M for K ions but further growth in the doping concentration, the flatband potential increased up to -0.87 V vs. Flatband potential that is more negative has the ability to assist the charge separation at the interface of the semiconductor/electrolyte [51]. It can be noted that the maximum flatband potential for 0.1M of Cu2O doped with K ions was observed, which is consistent with the maximum photocurrent density displayed by this sample [52]. The carrier concentration was calculated utilizing the following equation [50].

                                                            (7)    

where S is the plot’s slope of  Mott-Schottky and can be described as [50] :

                                                            (8)

Mott-Schottky plots for pure and K doped Cu2O exhibited a positive slope, signifying that all Cu2O thinfilms were n-type semiconductors. The donor density raised from 1.3 × 1017 to 3.2 × 1017 cm-3 with the increase in doping concentration, indicating a higher density of the vacancies in doped samples [53]. This may be due to two reasons: the first could be caused by substitution of Cu+2 in Cu2O by K+ and producing oxygen vacancies to maintain the charge neutrality [36]. The second reason may be due to that potassium acts as donor level and its incorporation in the crystal increases donor density [52, 54].

Table 2.  Influence of doping with K ions on carrier densities and flat band potentials for n-Cu2O thin films.

Samples

Vfb

(V vs. Ag/AgCl)

Conductivity type

Dopant’s density

(cm− 3)

Un doped Cu2O

-0.8

n-type

1.3×1017

0.03 K-Cu2O

-0.75

n-type

1.7×1017

0.05 K-Cu2O

-0.71

n-type

2.1×1017

0.07 K-Cu2O

-0.83

n-type

2.6×1017

0.1 K-Cu2O

-0.87

n-type

3.2×1017

Figure 9. Mott–Schottky measurements for K doped Cu2O with different concentrations.

  1. The photoelectrochemical measurements are too simple to demonstrate the corresponding properties of the system. The authors may need to add a full set of data to characterize the photoelectrochemical properties.

Thanks for such comment, and for that we add Electrochemical impedance spectroscopy (EIS) measurements as following:

Electrochemical impedance spectroscopy (EIS) measurements were utilized to identify the interfacial charge-transfer behavior for all samples in an electrolyte [49]. These measurements were made in 0.5 M Na2SO4 at a perturbation potential of 0.5 V  at the frequency range of 104 Hz. Figure 10 exhibits the Nyquist plot (Z imaginary vs. Z real) for Cu2O samples, which was fitted by a ZSimpWin software with the equivalent electrical circuit model of R(Q(R(Q(RW)))), including the solution-resistance (R1), charge-transfer resistance (R2), the adsorption resistance (R3), constant phase element (Q), Warburg’s impedance (W) [55]. The Nyquist plots include semicircles and its diameter equals the resistance to charge transfer (RCT) across the electrode/electrolyte interface. The more the arc diameter, the more the charge transfer resistance [56]. K ions doped Cu2O thin films and produced at 0.1 M are found to have a diameter that is significantly smaller than the diameters of the other semicircles, indicating faster charge transfer and lower rate of charge recombination, which is talented for p-Cu2O in homojunctions solar cells [36].

Figure 10. Nyquist plots (Z real versus Z imaginary) for undoped and K-doped n-Cu2O thin films measured in 0.5 M Na2SO4.

Round 2

Reviewer 2 Report

The authors have fully addressed my comments and I suggest to publish without further revisions.